# The Rochester Modified Magee Algorithm (RoMMa): An Outcomes Based Strategy for Clinical Risk-Assessment and Risk-Stratification in ER Positive, HER2 Negative Breast Cancer Patients Being Considered for Oncotype DX^®^ Testing

**DOI:** 10.3390/cancers15030903

**Published:** 2023-01-31

**Authors:** Bradley M. Turner, Brian S. Finkelman, David G. Hicks, Numbere Numbereye, Ioana Moisini, Ajay Dhakal, Kristin Skinner, Mary Ann G. Sanders, Xi Wang, Michelle Shayne, Linda Schiffhauer, Hani Katerji, Huina Zhang

**Affiliations:** 1Department of Pathology and Laboratory Medicine, University of Rochester Medical Center, 601 Elmwood Ave., Rochester, NY 14620, USA; 2M. Health Fairview Ridges, Burnsville, MN 55337, USA; 3Department of Medical Oncology, University of Rochester Medical Center, Rochester, NY 14642, USA; 4Department of Surgical Oncology, University of Rochester Medical Center, Rochester, NY 14642, USA; 5Norton Healthcare, University of Louisville Department of Pathology, Louisville, KY 40292, USA

**Keywords:** ER^+^ breast cancer, recurrence, average modified Magee score, algorithm, RoMMa, Oncotype DX^®^

## Abstract

**Simple Summary:**

As we move forward into the era of precision cancer medicine, we must consider the current state of the health care economy, as well patient access to quality health care. While multigene assays such as Oncotype DX^®^ have certainly allowed for a more individualized approach to the risk-stratification of estrogen receptor positive, HER2 negative breast cancer patients, multigene assays are costly (Oncotype DX^®^ costs more than USD $4000.00), and may not be accessible to breast cancer patients without adequate health insurance, particularly in more poverty stricken geographies around the world. These breast cancer patients also deserve opportunities for evidence-based clinical risk-assessment and risk-stratification. In this study we present data on the Rochester Modified Magee algorithm, a risk-stratification algorithm for breast cancer patients that could provide a significant cost savings for the health care system and also allow for clinical risk-assessment and risk-stratification when access to multigene testing is not readily available.

**Abstract:**

Introduction: Multigene genomic profiling has become the standard of care in the clinical risk-assessment and risk-stratification of ER^+^, HER2^−^ breast cancer (BC) patients, with Oncotype DX^®^ (ODX) emerging as the genomic profile test with the most support from the international community. The current state of the health care economy demands that cost-efficiency and access to testing must be considered when evaluating the clinical utility of multigene profile tests such as ODX. Several studies have suggested that certain lower risk patients can be identified more cost-efficiently than simply reflexing all ER^+^, HER2^−^ BC patients to ODX testing. The Magee equations^TM^ use standard histopathologic data in a set of multivariable models to estimate the ODX recurrence score. Our group published the first outcome data in 2019 on the Magee equations^TM^, using a modification of the Magee equations^TM^ combined with an algorithmic approach—the Rochester Modified Magee algorithm (RoMMa). There has since been limited published outcome data on the Magee equations^TM^. We present additional outcome data, with considerations of the TAILORx risk-stratification recommendations. Methods: 355 patients with an ODX recurrence score, and at least five years of follow-up or a BC recurrence were included in the study. All patients received either Tamoxifen or an aromatase inhibitor. None of the patients received adjuvant systemic chemotherapy. Results: There was no significant difference in the risk of recurrence in similar risk categories (very low risk, low risk, and high risk) between the average Modified Magee score and ODX recurrence score with the chi-square test of independence (*p* > 0.05) or log-rank test (*p* > 0.05). Using the RoMMa, we estimate that at least 17% of individuals can safely avoid ODX testing. Conclusion: Our study further reinforces that BC patients can be confidently stratified into lower and higher-risk recurrence groups using the Magee equations^TM^. The RoMMa can be helpful in the initial clinical risk-assessment and risk-stratification of BC patients, providing increased opportunities for cost savings in the health care system, and for clinical risk-assessment and risk-stratification in less-developed geographies where multigene testing might not be available.

## 1. Introduction

The evolution of precision medicine in breast cancer care increasingly requires the use of the most recent technologies. Multigene genomic profiling has become the standard of care in the clinical risk-assessment and risk-stratification of ER-positive, HER2-negative breast cancer patients, with Oncotype DX^®^ (ODX) emerging as the genomic profiling test with the most robust clinical data as well as the most support from the international community [1]. The clinical validation, prognostic benefit, and predictive value of ODX has been assessed for both lymph node negative and lymph node positive patients in at least 21 studies [1,2,3,4,5,6,7,8,9,10,11,12,13,14,15,16,17,18,19,20,21,22,23]. The clinical utility of ODX has been evaluated in at least 22 studies, for both lymph node–negative and lymph node–positive patients, including several prospective randomized clinical trials [1,2,24,25,26,27,28,29,30,31,32,33,34,35,36,37,38,39,40,41,42,43,44,45]. Neither the prognostic benefit nor the predictive value of ODX are in question.

There is no question that the clinical utility of a test should consider the quality of the associated health outcome [46,47]. However, clinical utility (the likelihood that a test will, by effectively guiding an intervention, result in an improved health outcome) is also inextricably associated with cost-effectiveness (cost-efficient production of an effective [health] outcome), and therefore, by definition, cost-efficiency (the least waste and best use of resources) should be considered when evaluating the clinical utility of any test, including multigene profile tests such as ODX. Published articles evaluating the cost-effectiveness of ODX have shown mixed results, with industry funded studies tending to show more favorable incremental cost-effectiveness ratios than non-industry funded studies [48,49]. ODX is an expensive test, and may not be the most cost-efficient option in certain subsets of breast cancer patients. 

Access to testing should also be considered when evaluating the clinical utility of testing methodologies, because a test only has clinical utility where it can be implemented into clinical practice. The mortality/incidence ratio of breast cancer is highest in less developed geographies, as high as 60% in middle Africa, with Melanesia and western Africa not far behind [1,50]. Access to multigene testing for breast cancer is not readily available in the less developed geographies of the world, and therefore, by definition, multigene testing provides no real clinical utility in these places, or to any breast cancer patient who does not have access to this testing methodology. Breast cancer patients without access to multigene testing should also have opportunities for treatment interventions based on evidence-based clinical risk-assessment and risk-stratification. Online risk prediction models, such as the ONCOassist breast adjuvant tool based on the PREDICT algorithm, are available, but these models generally incorporate relatively limited information on histopathologic variables [51,52,53,54]. Several studies have suggested that standard clinical, histopathologic, semi-quantitative immunohistochemistry (IHC), and biomarker data can provide information similar to that provided by the ODX recurrence score [13,32,49,55,56,57,58,59,60,61,62,63,64,65,66]. Multiple studies have also suggested that certain lower risk patients can be identified in a more cost-efficient manner than simply reflexing all ER-positive, HER2-negative breast cancer patients to ODX testing [49,66,67,68]. Risk-stratification models such as the IHC4 score [13] and the Magee equations^TM^ [49,55,56,65,66,67,68], both of which use semi-quantitative information from the immunohistochemical assessment of ER, PR, HER2, and Ki-67 (four of the genes measured in the ODX panel), have been validated for identifying patients at low, moderate, or high risk of breast cancer relapse following endocrine therapy [49,67,68,69]. An excellent review of the original and new Magee equations^TM^ has recently been published [70].

In 2019, our group published the first outcome data on the Magee equations^TM^, comparing a modification of the new Magee equation score, which we called the average Modified Magee score (aMMs), to the ODX recurrence score, suggesting that patients with an aMMs ≤12 had a comparably low risk of breast cancer recurrence when compared to the low-risk categories defined by ODX, in an average follow-up of 79 months (6.6 years) [49] A subsequent study by Bhargava et al. in 2020 reported that in patients who received only endocrine therapy, 100% of discordant cases with an expected ODX score of ≤25 by the Magee decision algorithm^TM^, but with an actual ODX score of >25, did not experience distant recurrence in an average follow-up of 73 months (6.1 years). There has been no published outcome data on the Magee equations^TM^ (either the original, new, or modified versions) since those two studies. In this study, we present additional outcome data on the average modified Magee score, with considerations of the ODX risk-stratification categories discussed in the TAILORx study [71].

## 2. Materials and Methods

Patients and data retrieval: A retrospective review of the pathology database to identify all patients at the University of Rochester who received ODX testing between 2007 and 2018 was performed. The pathology and medical databases were subsequently reviewed to further define patients considered to be eligible for this study. Eligibility criteria for this study included: (1) Received ODX testing; (2) ER-positive/HER2-negative invasive breast cancer; (3) Received either Tamoxifen or an aromatase inhibitor; (4) Did not receive any systemic chemotherapy; (5) Available information on ER/PR percentage and intensity, tumor size, tumor nuclear grade, tumor morphology, and tumor mitotic count per 10 high power fields; (6) ≥60 months of available follow-up without breast cancer recurrence; and, (7) any breast cancer recurrence. Lymph node involvement, the presence of lymphovascular invasion, Ki-67 status, age, and ethnic status were also included, if available. A final total of 355 patients were included in this study (Table 1).

All tumor H&E slides and IHC were reviewed by board-certified breast pathologists, with manual interpretation of ER (ERα clones ID5 and ER-2-123), PR (PR clone PgR1294] pharmDxTM), HER-2 (Polyclonal rabbit anti-human HER-2 HercepTestTM), and Ki-67 (Monoclonal mouse anti-human Ki-67 antigens [clone MIB-1, code M7240]). HER-2 FISH was performed (FDA-approved test kit [DAKO]—HER-2 IQFISH pharmDxTM) on all equivocal HER-2 IHC results to confirm a negative result. The Nottingham Score was calculated using the Nottingham modification of the Bloom-Richardson system [72]. Modified ER and PR H-scores were determined as described in the previously published literature [66]. Ki-67 was evaluated using a global estimate of the percentage of positive staining tumor cells rather than a hot spot estimate, as recommended by the International Ki-67 in Breast Cancer Working Group [73].

Study Design: The Rochester Modified Magee algorithmic approach has been previously described by Turner et al. [49,66]. Minor modifications, which update and simplify the previously published algorithmic approach, were made based on our current study results (Figure 1). Briefly, the aMMs was calculated [66]. All patients with an aMMs ≤12 were considered very low risk, all patients with an aMMs >12 and ≤ 18 were considered low risk, and all patients with an aMMs >18 were considered high risk. Patients were stratified into ODX similar risk categories with considerations of the results for risk-stratification discussed in the TAILORx study [71]. Specifically, all patients with an ODX of <11 were considered very low risk, all patients with an ODX of ≥11 and <16 were considered low risk, all patients >50 years of age with an ODX of ≥16 and ≤25 were also considered low risk, all patients ≤50 years of age with an ODX of ≥16 and ≤25 were considered high risk, and finally, all patients with an ODX >25 were also considered high risk. We compared the similar risk categories of the aMMs and ODX. 

Statistical analysis: Available clinical and pathologic data were summarized using percentages, descriptive statistics (mean, range, frequencies), and inferential statistics (chi-square [X^2^] test of independence, *t*-test). Kaplan-Meier plots were constructed and survival curves were compared using the log rank test. The sensitivity, specificity, positive-predictive value (PPV), and negative-predictive value (NPV) for predicting a high ODX score were calculated for strategies of avoiding ODX testing in patients with an aMMs below pre-specified cutoffs (≤12 and ≤18), as well as the proportion of individuals who would avoid ODX testing with each strategy. The area under the receiver operating characteristic (ROC) curve (AUC) was calculated separately using the sensitivity and specificity at each pre-specified cutoff, using the following formula: (sensitivity + specificity)/2 [74]. For the evaluation of these strategies, individuals above the cutoff (i.e., those who did not meet criteria to forego ODX testing) were assumed to have correct ODX results, as would have been obtained from actual ODX testing. All statistical analyses were performed using the statistical Analysis ToolPak (Microsoft Excel Office 2010 version 14.0.7015.100) except for the chi-square (X^2^) test of independence, which were performed using JavaStat 2-way Contingency Table Analysis (revised version 23 July 2013 http://statpages.org/ctab2×2.html [accessed on 25 January 2023]). For all results, a *p*-value of < 0.05 was considered significant. This study received IRB approval from the University of Rochester (IRB# RSRB00069270).

## 3. Results

Overall, 31/355 (8.7%) patients had a breast cancer recurrence (Table 1). There was no significant difference in the risk of recurrence in similar risk categories between the aMMs and ODX (Table 2) using the chi-square test of independence (*p* = 0.27 [very low risk]; *p* = 0.84 [low risk]; *p* = 0.83 [high risk]). There was no significant difference in disease free survival in similar risk categories between the aMMs and ODX (Figure 2A–C) using the log rank test (*p* = 0.995 [very low risk]; *p* = 0.999 [low risk]; *p* = 0.999 [high risk]). 

2/62 (3.2%) very low risk aMMs patients had a breast cancer recurrence, compared to 7/94 (7.4%) very low risk ODX patients (Table 2 and Table 3). A total of 14/173 (8.1%) low risk aMMs patients had a breast cancer recurrence, compared to 18/208 (8.7%) low risk ODX patients (Table 2 and Table 3). When considering all lower risk patients (very low risk and low risk), 16/235 (6.8%) lower risk aMMs patients had a breast cancer recurrence, compared to 25/302 (8.2%) lower risk ODX patients (Table 2 and Table 3).

A total of 6/62 (9.7%) very low risk aMMs patients were considered to be high risk by ODX (Table 4 and Appendix A). None (0%) of these discordant very low risk aMMs/high risk ODX patients had a breast cancer recurrence (Table 3). When considering all lower risk aMMs patients (very low risk and low risk), 25/235 (10.6%) lower risk aMMs patients were considered to be high risk by ODX (Table 4). Two out of 25 (8%) of these discordant lower risk aMMs/high risk ODX patients had a breast cancer recurrence (Table 3 and Appendix A).

A total of 15/94 (16%) very low risk ODX patients were considered to be high risk by the aMMs (Table 4 and Appendix A). Two out of 15 (13.3%) of these discordant very low risk ODX/high risk aMMs patients had a breast cancer recurrence (Table 3 and Appendix A). When considering all lower risk ODX patients (very low risk and low risk), 92/302 (31%) lower risk ODX patients were high risk by the aMMs (Table 4), and 11/92 (12%) of these discordant lower risk ODX/high risk aMMs patients had a breast cancer recurrence (Table 3 and Appendix A).

The strategies of foregoing ODX testing in patients whose aMMs was below pre-specified cutoffs (≤12 and ≤18) were also assessed for their ability to predict having a high-risk ODX. For the strategy of foregoing ODX testing in patients with a very low aMMs (≤12), the AUC was 0.94, with a sensitivity of 89%, a specificity of 100%, a PPV of 100%, and a NPV of 98%. In this strategy, approximately 17% of patients would be able to avoid ODX testing. The performance of the aMMs for predicting high ODX decreases when combining very low or low categories together (using a cutoff of an aMMs ≤ 18), which would allow approximately 66% of patients to avoid ODX testing. In this strategy, the AUC for predicting high ODX was 0.76, with a sensitivity of 53%, a specificity of 100%, a PPV of 100%, and an NPV of 92%. 

## 4. Discussion

The evolution of precision medicine will continue to involve the integration of genomic and molecular technologies into the diagnostic and treatment algorithms of patients. While genomic and molecular technologies clearly have value, their clinical utility to the health care system is compromised if another test can be shown to be equally effective but less expensive. It is critical that considerations surrounding genomic testing be interpreted within the morphologic and clinical context for each patient. Therefore, continued research on less expensive methods, such as immunohistochemistry-based algorithms, is important. Challenges remain in accurately identifying which strategies are more cost-effective and cost-efficient in identifying unique subsets of breast carcinoma patients who may benefit from systemic chemotherapy.

Currently, several multigene expression-based assays which offer prognostic and predictive values in certain subgroups of ER-positive breast cancers have been established and made commercially available, including ODX, MammaPrint (Agendia, Amsterdam, The Netherlands), Prosigna (NanoString Technologies Inc. Seattle, WA, USA), EndoPredict (Sividon Diagnostics GmbH, Holn, Germany), and Breast Cancer Index (Biotheranostic, San Diego, CA, USA) [1,49,75,76,77]. These molecular-based assays are expensive, and access to the test systems vary from country to country. Of these genomic assays, ODX has perhaps the most robust clinical data as well as the most support from the international community [1]. The ODX test costs over USD $4000, and the cost-effectiveness of ODX has been supported in the published literature based on the reduction in use of adjuvant chemotherapy [49,78,79,80,81,82,83,84,85,86] in patients with low-risk ODX results. However, many of these studies were industry funded and incorporated study designs that might increase the risk of bias, such as (1) constructed simulations of practice that may not reflect actual practice; (2) model structures ignoring clinicopathologic information and/or combining different risk groups; (3) model assumptions that ignore chemotherapy toxicity, assume the predictive value of ODX, and/or assume how often ODX is used in certain populations; and, (4) input parameter selections that may not reflect real-world population distributions, such as implausible estimates of chemotherapy effectiveness and assumptions about age at breast cancer diagnosis [48,49,78]. Our group previously suggested a potential estimated cost savings to the health-care system in 2018 of over USD $100,000,000 if ODX testing was avoided in certain subsets of breast cancer patients [49]. Other authors have similarly suggested a potential for substantial cost savings to the health-care system if ODX testing was avoided in certain subsets of breast cancer patients [66,67,78,87]. For instance, a recent study based on real-world population based data by Mittmann et al. suggests that using ODX to determine whether adjuvant chemotherapy should be added to endocrine therapy in ER-positive lymph node–negative breast cancer patients was approximately USD $3000 more expensive per patient than not using the test [49,78]. There remains significant uncertainty on the part of many clinicians on how to best integrate the ODX results with the available clinicopathologic features of a patient’s tumor in certain subsets of breast cancer patients [48,49,88].

Four of the 16 cancer related genes measured by ODX include those that express ER, PR, HER2, and Ki-67. The protein expression of ER, PR, and HER2 is routinely assessed by IHC as part of the diagnostic evaluation of breast cancer, and many institutions also routinely assess Ki-67. Several studies have been published proposing models that can accurately predict ODX low- and high-risk categories using clinical and pathological variables such as tumor morphology, tumor architecture, nuclear grade, mitotic count, ER, PR, HER2, and Ki-67 [49,51,52,61,62,63,64,89,90]. The original and new Magee equations^TM^ use a linear regression model, originally developed at the University of Pittsburgh Medical Center Magee Women’s Hospital, that combines clinical and pathological data to estimate the ODX recurrence score [51,52]. Our group and others have subsequently published data supporting the use of a modification of the new Magee equations developed at the University of Rochester Medical Center (a modification developed to make the equations easier to use), in an algorithmic step-wise approach, to risk stratify patients into lower and higher risk of breast cancer recurrence [49,62,91,92,93,94].

In 2019, our group revised the initial algorithmic approach [66] and published the Rochester Modified Magee algorithm (RoMMa), [49] with the first outcome data on the Magee equations^TM^, suggesting that patients with an aMMs of ≤12 had a comparably low risk of breast cancer recurrence when compared to the low-risk categories defined by ODX [49,66]. In that 2019 study, which had an average follow-up of 83 months in 55 patients with an aMMs ≤12, only 1 of these 55 patients (1.8%) had a breast cancer recurrence. Only one other study addressing outcomes associated with the Magee equations^TM^ has been published since our 2019 study [67]; however, this study by Bhargava et al. in 2020 only reported the outcomes of 26 patients classified as lower risk by the Magee decision algorithm^TM^ that had a discordant higher risk ODX score of >25. In that study, none of these 26 discordant cases experienced distant recurrence (average follow-up of 73 months). There has been no published outcome data on the Magee equations^TM^ (either the original, new, or modified versions) since the 2019 Turner et al. [49] and 2020 Bhargava et al. [67] studies. 

The risk-stratification of breast cancer patients based on the ODX recurrence score has evolved over the last several years, culminating with results from the TAILORx trial, [71] which suggests that breast cancer patients with an ODX < 16, and breast cancer patients >50 years of age with an ODX ≤ 25, have a lower risk of breast cancer recurrence, and may not benefit from adjuvant systemic chemotherapy. In our 2019 study [49], 12/258 (4.7%) patients with an ODX ≤ 25 had a breast cancer recurrence, consistent with the results from the TAILORx trial [71]. In our 2019 study, 537 cases had an aMMs ≤ 18, [49] and 98% of these 537 cases had an ODX recurrence score ≤ 25. This finding was supported in a 2019 study by Bhargava et.al., who reported that 98% of cases defined as lower risk by the Magee decision algorithm^TM^ (which included all cases with an average Magee score < 18), had an ODX ≤ 25 [68]. Based on the risk-stratification categories supported by our 2019 study [49], the Bhargava et al. studies, [67,68] and the TAILORx trial, [71] in the current study we defined a very low-risk group, a low-risk group, and a high-risk group (Table 2), and compared similar aMMs and ODX risk-stratification categories.

Our results did not find any significant difference in similar risk categories between the aMMs and ODX using the chi-square test of independence (Table 2) or Kaplan–Meier analysis (Figure 2A–C). Of interest, although the difference was not significant, in our study, lower risk aMMs patients actually had a lower point estimate of risk of breast cancer recurrence compared to lower risk ODX patients (Table 2, Figure 2A,B). 

When considering discordant patients, none our very low risk aMMs patients considered to be high risk by ODX (Table 4 and Appendix A) had a breast cancer recurrence (Table 3). These very low risk aMMs/high risk ODX patients had relatively lower grade, relatively higher ER and PR H-scores, and relatively lower Ki-67 (Appendix A, Figure 3A,B). When considering all lower risk aMMs patients (very low and low risk), 25/235 (10.6%) were considered to be higher risk by ODX (Table 4), with 2 of these 25 discordant low-risk aMMs/high risk ODX patients (8%) having a breast cancer recurrence (Table 3 and Appendix A). Notably, both of these low-risk aMMs/high risk ODX patients with a breast cancer recurrence were ≤50 years of age, with an ODX between 16 and 25. When examining the histology of these two discordant low-risk aMMs/high-risk ODX patients who recurred, both were relatively lower to intermediate grade, (Figure 4A,B), with relatively higher ER H-scores (Appendix A); however, one of these two recurrent patients had a relatively low PR H-score, and Ki-67 was not available in either patient, which may have been helpful in deciding whether or not to offer systemic chemotherapy.

When considering discordant patients with a very low-risk ODX and a high-risk aMMs, 15/94 (16%) very low risk ODX patients were considered to be high risk by the aMMs (Table 4 and Appendix A). These patients typically had at least intermediate grade tumors (Figure 5), with relatively higher Ki-67 percentages of at least 10%, relatively lower PR H-scores, and occasionally with relatively lower ER-H scores (Appendix A). A total of 2 of these 15 discordant very low-risk ODX/high-risk aMMs patients (13.3%) had a breast cancer recurrence (Table 3 and Appendix A). When considering all lower-risk ODX patients (very low and low-risk), 92/302 (12%) lower-risk ODX patients were considered to be higher-risk by the aMMs (Table 4), with 11 of these 92 lower-risk ODX/high-risk aMMs discordant patients (12%) having a breast cancer recurrence (Table 3 and Appendix A). When examining the histology of these eleven lower-risk ODX/high-risk aMMs patients that recurred, they typically had at least intermediate grade tumors (Figure 6), often with higher Ki-67 percentages of at least 20%, relatively lower PR H-scores, and occasionally relatively lower ER-H scores (Appendix A).

The data on discordance between the ODX recurrence score and clinocopathologic factors is limited; however, the available literature does suggest that there are patients assessed as low-risk by clinocopathologic variables that have had discordant high risk ODX scores [22,60,95,96]. The reasons for this are not entirely clear; however, the algorithm ODX uses to calculate the recurrence score gives the highest weight to proliferation (which includes Ki-67), and it has been shown that a proliferating, cellular stroma and/or admixed inflammatory cells may result in an artificially increased ODX recurrence score in low-grade invasive breast cancers [95]. The reasons why a patient with high-risk clinocopathologic variables would have a discordant low risk ODX recurrence score remain unclear, and warrants further investigation. If there is a significant discordance between the ODX recurrence score and clinocopathologic variables, this should be thoroughly investigated (i.e., the slide of the block selected for the multigene assay should be reviewed, and consideration for sending another block should be given) before decisions on adjuvant chemotherapy are made.

Earlier studies have demonstrated that the patients with a lower aMMs are likely to have a lower ODX [49,51,52,66,67,68,91,92,93,94] with similar recurrence outcomes [49]. In this study, there was no significant difference in the risk of recurrence in similar risk categories (very low-risk, low-risk, and high-risk) between the aMMs and ODX recurrence score with the chi-square test of independence (*p* > 0.05) or log-rank test (*p* > 0.05). In addition, the performance of the aMMs in this modified Magee Equation model for predicting breast cases with low ODX is good, especially in the very low category. These findings continue to strongly support that there would be high value in developing a “triage” strategy when considering which patients to send out for ODX testing. As illustrated in Figure 1, in patients who have an aMMs ≤ 12 (very low-risk, ~17% of our study population), ODX testing will likely not provide additional clinical utility, and should only be considered with discordant clinical and/or pathologic features. In patients with an aMMs >12 and ≤18 (low-risk, ~49% of our study population), given decreased sensitivity for excluding high ODX, ODX testing may or may not provide additional clinical utility, and testing should be considered in conjunction with clinical and/or pathologic features. On the contrary, although patients who are higher risk by the aMMs also had similar recurrence outcomes compared to patients who are higher risk by ODX, randomized studies evaluating the aMMs and the efficacy of systemic chemotherapy would likely be necessary before any definitive conclusions as to whether or not patients with a higher risk aMMs could forgo ODX testing and be considered for systemic chemotherapy treatment.

As mentioned earlier, several models have been proposed that can accurately predict ODX low- and high-risk categories [49,51,52,61,62,63,64,89,90], and some of these models also used routinely-reported pathological variables such as tumor morphology, tumor architecture, nuclear grade, mitotic count, ER, PR, HER2, and Ki-67 [49,51,52,61,62,63,64,89,90]. For example, Marazzi et al. generated a statistical model based on the quantitative IHC results for ER, PR, and Ki-67. It demonstrated a good performance of this statistical model for predicting an ODX recurrence score of ≤25, with an AUC of 0.922 and 0.823, a sensitivity of 84.1% and 86.2%, and a specificity of 80% and 56.6% in their internal and external validation cohorts, respectively [90]. In our study, using the RoMMa and based on the average modified Magee score at a cutoff of ≤12, variables including ER, PR, Ki-67, as well as Nottingham score and tumor size were included and achieved an AUC of 0.94, with a sensitivity of 89%, a specificity of 100%, a PPV of 100%, and a NPV of 98% for excluding high ODX in the cases with a very low aMMs. This algorithm would avoid ODX testing in approximately 17% of patients, based on our pre-specified cutoff and the cohort included in our study. Additional research examining different aMMs cutoffs as well as different cutoffs in patients above and below 50 years of age is warranted to see if a larger group of patients can potentially safely forego ODX testing. Additional research examining the accuracy of the aMMs and the RoMMa in the larger population of ER-positive, HER2-negative breast cancer patients, including those receiving chemotherapy is also warranted. However, based on our current results, the RoMMa shows promise to serve as a strategy for risk-stratifying breast cancer patients with the potential to provide significant cost savings for the health care system by identifying a subgroup of patients who can safely forego ODX testing, and also to allow for evidence-based clinical risk-assessment and risk-stratification when access to multigene testing is not available.

Our study does have some limitations. We do not have data on adherence to hormonal and radiation therapies in our cohort, which might affect outcomes; however, we believe that therapy adherence is unlikely to be different across risk categories and therefore less likely to bias our results. Access to larger database populations such as ECOG, NSABP-14, NSABP B-20, and SEER-Medicare would be helpful in providing additional longitudinal data with 10 or more years of follow-up, which we believe will further validate our findings [49]. The histologic grading and immunohistochemical reporting in our study was based on the original diagnostic pathology report, potentially leading to concerns with poorer reproducibility and accuracy compared to central pathology review. The literature supports the reproducibility of ER, PR, and HER2, [97,98] with well-standardized guidelines available to guide practicing pathologists in their interpretation. Less consensus exists on the reproducibility and best approach to the evaluation of Ki-67, although standardized criteria have been proposed by the International Ki-67 Working Group [68]. All cases in our study were evaluated by pathologists with subspecialty training in breast pathology, which we believe minimizes problems with reproducibility in the interpretation of histopathologic and immunohistochemical variables. Furthermore, if algorithms such as the RoMMa are to prove useful in clinical practice, they will need to be based on data provided by practicing pathologists. Therefore, we believe that our approach helps to improve the external validity of our proposed algorithm.

## 5. Conclusions

Our results continue to support that those patients with a lower risk aMMs, who are lower-risk by RoMMa criteria, are likely to have a lower-risk ODX recurrence score. These lower risk patients by the RoMMa also have a lower risk of breast cancer recurrence that is comparable to patients with a lower risk ODX recurrence score. The risk-stratification of breast cancer patients should begin with the evaluation of available clinical and pathologic metrics, and the aMMs can be helpful in such evaluations. Consideration should be given to not sending out tissue for ODX testing in those patients who have a lower-risk aMMs and are lower-risk by RoMMa criteria, particularly if the aMMs is ≤12. ODX testing should only be done on patient tissue where the assay results would potentially provide clinical utility beyond the available clinical and pathologic metrics. This kind of “triage” approach when risk stratifying breast cancer patients could provide a significant cost savings for the health care system and also allow for evidence-based clinical risk-assessment and risk-stratification when access to multigene testing is not available.

## Figures and Tables

**Figure 1 cancers-15-00903-f001:**
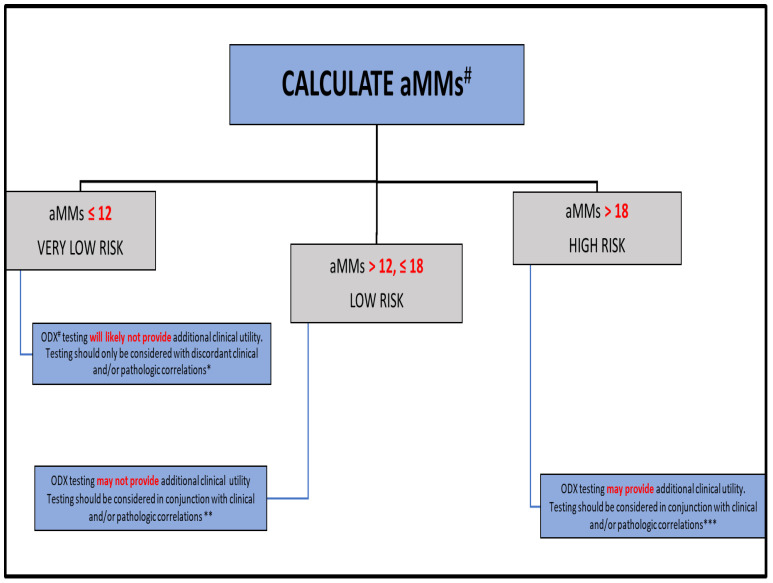
Rochester Modified Magee algorithm (RoMMa). # average Modified Magee score (aMMs); Oncotype DX^®^score (ODX). * Patients with an aMMs ≤ 12 will likely have a <5% risk of breast cancer recurrence, comparable to patients with an ODX < 11. ** Patients with an aMMs between 12 and 18 may have a 5–8% risk of breast cancer recurrence, comparable to lower risk patients with an ODX between 11 and 25. *** Patients with an aMMs >18 may have a >10% risk of breast cancer recurrence, comparable to higher risk ODX patients.

**Figure 2 cancers-15-00903-f002:**
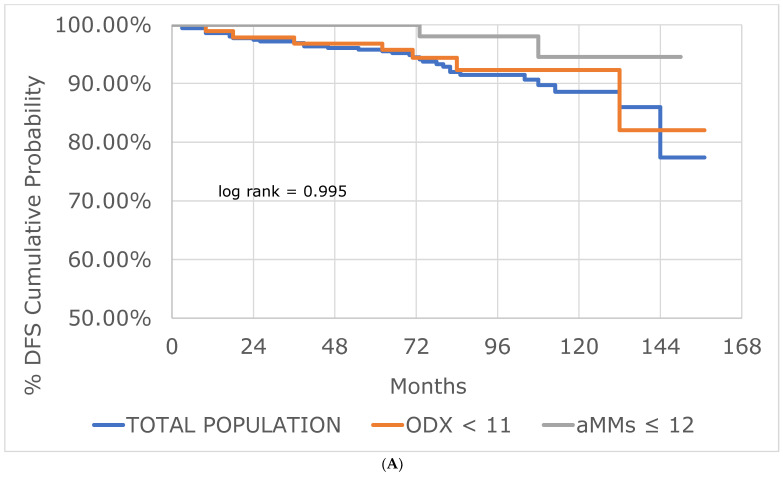
Kaplan–Meier curves, disease free survival (DFS) in the very low risk (**A**), low risk (**B**), and high risk (**C**) breast cancer populations. Abbreviations: aMMs = average Modified Magee score; ODX = Oncotype DX^®^.

**Figure 3 cancers-15-00903-f003:**
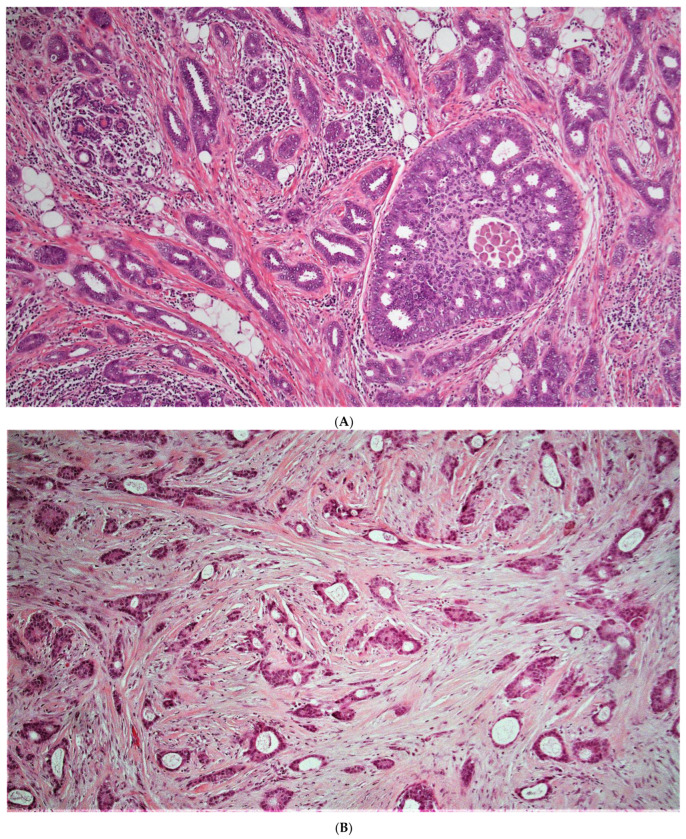
(**A**,**B**) (10×): Two examples of very low risk aMMs patients considered to be higher risk by ODX (**A**) ([Appendix A—case # 4]; (**B**) [Appendix A—case # 6]). Note the low grade histologic qualities in both patients. Note the background inflammation and DCIS in (**A**), both of which have been suggested as potential artifacts associated with an elevated ODX recurrence score [95]. Note the somewhat cellular stroma in (**B**), which has also been associated with an elevated ODX recurrence score [95]. Neither of these patients recurred (follow-up of 98 and 137 months, respectively).

**Figure 4 cancers-15-00903-f004:**
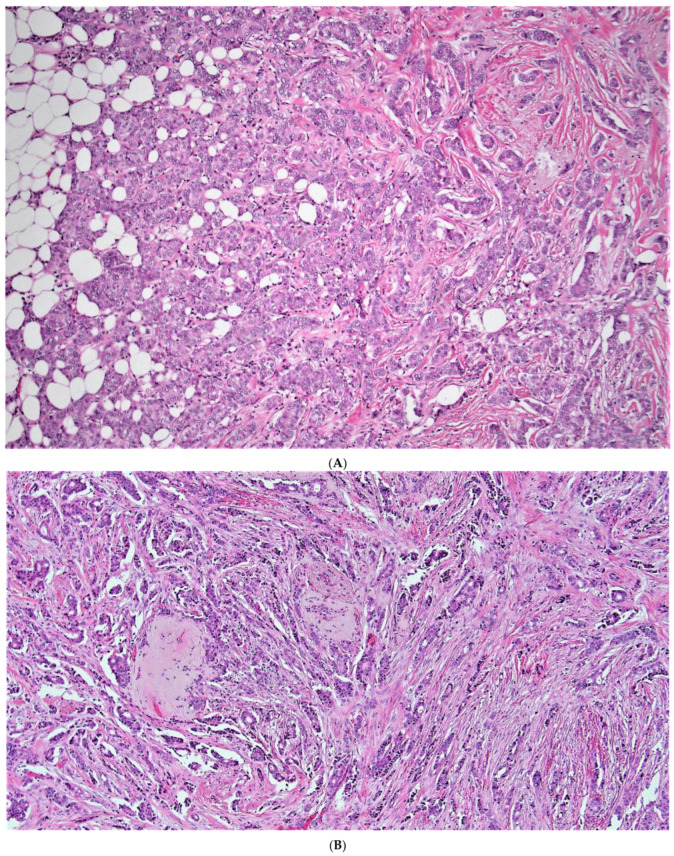
(**A**,**B**) (10×): The two low risk aMMs patients considered to be high risk by ODX with a breast cancer recurrence (**A**) ([Appendix A—case # 1]; (**B**) [Appendix A—case # 2]). Note the lower grade (**A**) to intermediate grade (**B**) histologic qualities in both patients. As in (**A**) and (**B**), note the background inflammation in (**A**), and the somewhat cellular stroma in (**B**), both of which have been associated with an elevated ODX recurrence score [95].

**Figure 5 cancers-15-00903-f005:**
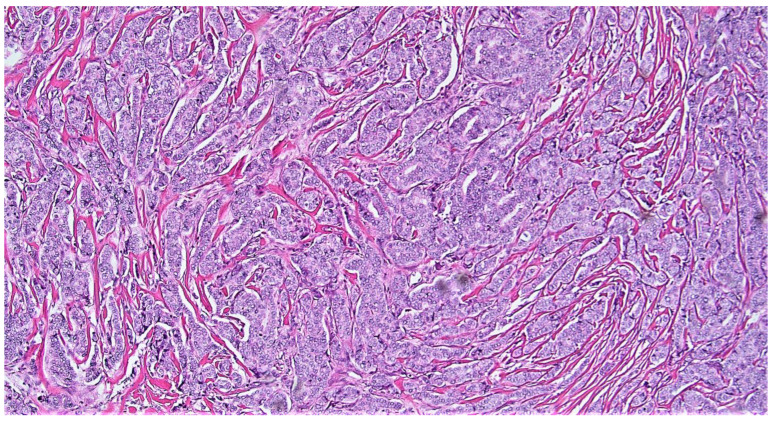
(10×): A very low-risk ODX patient considered to be high-risk by the aMMs with a breast cancer recurrence (Appendix A—case # 1). Note the intermediate grade histologic qualities. This patient was 69 years of age with an aMMs of 19.2, a Ki-67 of 30%, and an ODX recurrence score of 6.

**Figure 6 cancers-15-00903-f006:**
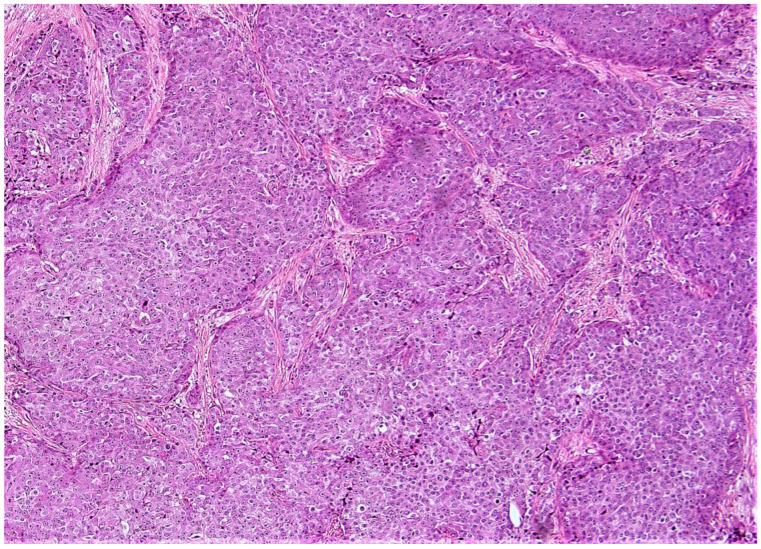
(10×): Example of a low-risk ODX patient considered to be higher-risk by the aMMs, with a breast cancer recurrence (Appendix A—case # 3) Note the higher grade histologic qualities. This patient was 67 years of age with an aMMs of 21.1, a Ki-67 of 15%, and an ODX recurrence score of 23.

**Table 1 cancers-15-00903-t001:** Demographics.

	N (%)	Mean Months of Follow-Up (Range)
TOTAL POPULATION	355 (100)	93.2 (20–60)
White	310 (87.3)	92.8 (20–160)
Black	32 (9.0)	90.7 (62–41)
Hispanic	6 (1.7)	117.3 (91–146)
Asian	5 (1.4)	107 (96–119)
Unknown	2 (0.6)	94 (90–98)
aMMs * ≤ 12	62 (17.5)	102.2 (63–150)
aMMs >12 & ≤18	173 (48.7)	94.3 (59–160)
aMMs > 18	120 (33.8)	87.1 (20–142)
ODX * < 11	94 (26.5)	91.9 (62–157)
ODX 11–25 **	208 (58.6)	93.9 (24–154)
ODX > 25 **	53 (14.9)	90.1 (20–160)
RECURRENCES	31 (8.7)	59.3 (3–144) ***
	MEAN (RANGE)	
AGE	59 (33–84)	
AGE OF RECURRENCE	65 (43–83)	
aMMs	16.7 (6.5–32.3)	
ODX	14.6 (0–44)	
NS *	5.7 (3–9)	
ER-H SCORE ****	261.1 (40–300)	
PR-H SCORE ****	175.0 (0–300)	
Ki-67	14.0 (0–70)	
Size (cm)	2.2 (0.4–14)	

* aMMs = average Modified Magee score; ODX = Oncotype DX; NS = Nottingham score. ** All patients with an ODX 16–25 and ≤50 years of age are included in the ODX >25 group. *** Follow-up until recurrence. **** Modified H-score [66].

**Table 2 cancers-15-00903-t002:** Risk categories and outcome.

	OUTCOME		
Risk Category	Recurrence N (%)	No Recurrence N (%)	*p*-Value
VERY LOW (N)			
Average Modified Magee score ≤12 (62)	2 (3.2)	60 (96.8)	0.27
Oncotype DX <11 (94)	7 (7.4)	87 (92.6)	
LOW (N)			
Average Modified Magee score >12, ≤18 (173)	14 (8.1)	159 (91.9)	0.84
Oncotype DX 11–25 (208) *	18 (8.7)	190 (91.3)	
HIGH (N)			
Average Modified Magee score >18 (120)	15(12.5)	105 (87.5)	0.83
Oncotype DX ≥16–25 (32) ** and Oncotype DX >25 (21) ***	6 (11.3)	47 (88.7)	

* Does not include patients ≤50 years of age with an Oncotype DX score of ≥16–25. ** Patients ≤50 years of age with an Oncotype DX score of ≥16–25 (n = 2 recurred). *** All patients with an Oncotype DX of >25.

**Table 3 cancers-15-00903-t003:** Recurrences and associated risk categories.

	ODX Risk Category **			
aMMs Risk Category *	Very Low	Low	High	TOTAL
Very low	0	2	0	2
Low	5	7	2 ***	14
High	2	9	4	15
**TOTAL**	7	18	6	31

* aMMs = average Modified Magee score; very low = aMMs ≤12; low = aMMs >12 & ≤18; high = aMMs >18. ** ODX = Oncotype DX; very low = ODX <11; low = ODX 11–25 (does not include patients with an ODX 16–25 and ≤ 50 years of age); high = ODX >25 (includes patients with an ODX 16–25 and ≤50 years of age [n = 2]). *** Both patients with an ODX 16–25 and ≤50 years of age.

**Table 4 cancers-15-00903-t004:** Total population and associated risk categories.

	ODX Risk Category **			
aMMs Risk Category *	Very Low	Low	High	TOTAL
Very low	26	30	6	62
Low	53	101	19 ***	173
High	15	77	28	120
**TOTAL**	94	208	53	355

* aMMs = average Modified Magee score; very low = aMMs ≤12; very low = aMMs >12 & ≤18; high = aMMs >18. ** ODX = Oncotype DX; very low = ODX <11; low = ODX 11–25 (does not include patients with an ODX 16–25 and ≤50 years of age); high = ODX >25 (includes patients with an ODX 16–25 and ≤50 years of age [n = 32]). *** All 19 patients are ≤50 years of age with an ODX 16–25.

## Data Availability

The data presented in this study are available on request from the corresponding author. The data are not publicly available due to privacy reasons.

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
