# Peer review of "The Rochester Modified Magee Algorithm (RoMMa): An Outcomes Based Strategy for Clinical Risk-Assessment and Risk-Stratification in ER Positive, HER2 Negative Breast Cancer Patients Being Considered for Oncotype DX® Testing"

_cancers, 2023, doi:10.3390/cancers15030903_

Round 1
Reviewer 1 Report
The manuscript is about RoMMA algorithm developed to predict ODX score, applied on a retrospective cohort.
Major comments
- Please give more details about negative or positive predictive value of the model and AUC results
- The discussion paragraph need more detailed guide to clinical appication of this predictive score
- In literature other similar systems are published to predict ODX results. For example, ADAPTED01 by Marazzi F. et al. A brief literature review to compare different systems for predicting ODX would make the manuscript even more interesting.
Author Response
We thank the reviewer for the opportunity to revise and resubmit our manuscript. We have made revisions with the goal of addressing the reviewers’ comments to the best of our ability. Responses to specific reviewer comments are in italics below.
Reviewer#1
The manuscript is about RoMMA algorithm developed to predict ODX score, applied on a retrospective cohort.
Major comments
- Please give more details about negative or positive predictive value of the model and AUC results
We thank the reviewer for the opportunity to further elaborate on our findings.This comment has been addressed by adding the results of performance of aMMs model for predication of ODX, including sensitivity, specificity, PPV and NPV (pages 5, 10, and 17).
- The discussion paragraph need more detailed guide to clinical application of this predictive score
We thank the reviewer for the recognition of this. This comment has been addressed by further describing the clinical application of this predicative score in the Discussion (pages 16 and 17)
- In literature other similar systems are published to predict ODX results. For example, ADAPTED01 by Marazzi F. et al. A brief literature review to compare different systems for predicting ODX would make the manuscript even more interesting.
We thank the reviewer for the acknowledgment of this fact. This comment has been addressed by including a paragraph which briefly discussed the current efforts on developing models to predict ODX (page 11 and 17).
Reviewer 2 Report
Excellent and very original manuscript. Following reference should also be included: Dannehl D, Engler T, Volmer LL, Staebler A, Fischer AK, Weiss M, Hahn M, Walter CB, Grischke EM, Fend F, Taran FA, Brucker SY, Hartkopf AD. Recurrence Score® Result Impacts Treatment Decisions in Hormone Receptor-Positive, HER2-Negative Patients with Early Breast Cancer in a Real-World Setting-Results of the IRMA Trial. Cancers (Basel). 2022 Oct 31;14(21):5365. doi: 10.3390/cancers14215365.
Author Response
We thank the reviewer for the opportunity to revise and resubmit our manuscript. We have made revisions with the goal of addressing the reviewers’ comments to the best of our ability. Responses to specific reviewer comments are in italics below.
Reviewer#2
Excellent and very original manuscript. Following reference should also be included: Dannehl D, Engler T, Volmer LL, Staebler A, Fischer AK, Weiss M, Hahn M, Walter CB, Grischke EM, Fend F, Taran FA, Brucker SY, Hartkopf AD. Recurrence Score® Result Impacts Treatment Decisions in Hormone Receptor-Positive, HER2-Negative Patients with Early Breast Cancer in a Real-World Setting-Results of the IRMA Trial. Cancers (Basel). 2022 Oct 31;14(21):5365. doi: 10.3390/cancers14215365.
We thank the reviewer for the acknowledgment of our study. This comment has been addressed by including the suggested reference (page 10)
Reviewer 3 Report
The authors report additional data on the usefulness of the Magee equations to predict (and avoid) Oncotype DX in certain low-risk groups of breast cancer patients. The data are of interest for many breast cancer oncologists, especially in low-income countries. The authors conclude that Oncotype DX is the test system with the most support from the international community. This might be questioned, however, other major test sytems like the Prosigna test etc. should at least be mentioned and discussed in the manuscript. In addition, access to test systems may vary from country to country. Many countries in Europe provide multigene genomic profiling at no cost to all of their breast cancer patients.
Author Response
We thank the reviewer for the opportunity to revise and resubmit our manuscript. We have made revisions with the goal of addressing the reviewers’ comments to the best of our ability. Responses to specific reviewer comments are in italics below.
Reviewer#3
The authors report additional data on the usefulness of the Magee equations to predict (and avoid) Oncotype DX in certain low-risk groups of breast cancer patients. The data are of interest for many breast cancer oncologists, especially in low-income countries. The authors conclude that Oncotype DX is the test system with the most support from the international community. This might be questioned, however, other major test sytems like the Prosigna test etc. should at least be mentioned and discussed in the manuscript. In addition, access to test systems may vary from country to country. Many countries in Europe provide multigene genomic profiling at no cost to all of their breast cancer patients.
We thank the reviewer for the acknowledgment of these facts. This comment has been addressed by mentioning other genomic assays in breast cancer including MammaPrint (Agendia, Amsterdam, The netherland), Prosigna (NanoString Technologies Inc. Seattle, WA, USA), EndoPredict (Sividon Diagnostics GmbH, Holn, Germany), and Breast Cancer Index (Biotheranostic, San Diego, CA, USA). In addition, we mentioned that these molecular-based assays are expensive, and access to the test system vary from country to country (page 10)
Round 2
Reviewer 1 Report
The authors performed an extensive review based on comments of review. I think that the manuscript can be considered for publication in the present form.